# Equivariant $Q$ Learning in Spatial Action Spaces

**Dian Wang**      **Robin Walters**      **Xupeng Zhu**      **Robert Platt**
Khoury College of Computer Sciences
Northeastern University
United States
{wang.dian, r.walters, zhu.xup, r.platt}@northeastern.edu

**Abstract:** Recently, a variety of new equivariant neural network model architectures have been proposed that generalize better over rotational and reflectional symmetries than standard models. These models are relevant to robotics because many robotics problems can be expressed in a rotationally symmetric way. This paper focuses on equivariance over a visual state space and a spatial action space – the setting where the robot action space includes a subset of $\mathrm{SE}(2)$. In this situation, we know a priori that rotations and translations in the state image should result in the same rotations and translations in the spatial action dimensions of the optimal policy. Therefore, we can use equivariant model architectures to make $Q$ learning more sample efficient. This paper identifies when the optimal $Q$ function is equivariant and proposes $Q$ network architectures for this setting. We show experimentally that this approach outperforms standard methods in a set of challenging manipulation problems.

**Keywords:** Reinforcement Learning, Equivariance, Manipulation

## 1   Introduction

A key question in policy learning for robotics is how to leverage structure present in the robot and the world to improve learning. This paper focuses on a fundamental type of structure present in visuo-motor policy learning for most robotics problems: translational and rotational invariance with respect to camera viewpoint. Specifically, the reward and transition dynamics of most robotics problems can be expressed in a way that is invariant with respect to the camera viewpoint from which the agent observes the scene. In spite of the above, most visuo-motor policy learning agents do not leverage this invariance in camera viewpoint. The agent's value function or policy typically considers different perspectives on the same scene to be different world states. A popular way to combat this problem is through visual data augmentation, i.e., to create additional samples or experiences by randomly translating and rotating observed images [1] but keeping the same labels. This can be used in conjunction with a contrastive term in the loss function which helps the system learn an invariant latent representation [2, 3]. While these methods can improve generalization, they require the neural network to learn translational and rotational invariance from the augmented data.

Our key idea in this paper is to model rotational and translation invariance in policy learning using neural network model architectures that are equivariant over finite subgroups of $\mathrm{SE}(2)$. These equivariant model architectures reduce the number of free parameters using steerable convolutional layers [4]. Compared with traditional methods, this approach creates an inductive bias that can significantly improve the sample efficiency of the model, the number of environmental steps needed to learn a policy. Moreover, it enables us to generalize in a very precise way: everything learned with respect to one camera viewpoint is automatically also represented in other camera perspectives via selectively tied parameters in the model architecture. We focus our work on $Q$ learning in spatial action spaces, where the agent's action space spans $\mathrm{SE}(2)$ or $\mathrm{SE}(3)$. We make the following contributions. First, we identify the conditions under which the optimal $Q$ function is $\mathrm{SE}(2)$ equivariant. Second, we propose neural network model architectures that encode $\mathrm{SE}(2)$ equivariance in the $Q$ function. Third, since most policy learning problems are only equivariant in *some* of the state variables, we propose partially equivariant model architectures that can accommodate this. Finally, we compare equivariant models against non-equivariant counterparts in the context of several robotic

5th Conference on Robot Learning (CoRL 2021), London, UK.

manipulation problems. The results show that equivariant models are more sample efficient than non-equivariant models, often by a significant margin. Supplementary video and code are available at https://pointw.github.io/equi_q_page.

## 2 Related Work

Data Augmentation: Data augmentation techniques have long been employed in computer vision to encode the invariance property of translation and reflection into neural networks [5, 6]. Recent work demonstrates the use of data augmentation improves the data efficiency and the policy's performance in reinforcement learning [7, 8, 9]. In the context of robotics, data augmentation is often used to generate additional samples [1, 10, 11]. In contrast to learning the equivariance property using data augmentation, our work utilizes the equivariant network to hard code the symmetries in the structure of the network to achieve better sample efficiency.

Contrastive Learning: Another approach to learning a representation that is invariant to translation and rotation is to add a contrastive learning term to the loss function [2]. This idea has been applied to reinforcement learning in general [3] and robotic manipulation in particular [12]. While this approach can help the agent learn an invariant encoding of the data, it does not necessarily improve the sample efficiency of policy learning.

Equivariant Learning: Equivariant model architectures hard-code E(2) symmetries into the structure of the neural network and have been shown to be useful in computer vision [13, 4, 14]. In reinforcement learning, some recent work applies equivariant models to structure-finding problems involving MDP homomorphisms [15, 16]. In addition, Mondal et al. [17] recently applied an E(2)-equivariant model to $Q$ learning in an Atari game domain, but showed limited improvement. To our knowledge, equivariant model architectures have not been explored in the context of robotics applications.

Spatial Action Representations: Several researchers have applied policy learning in spatial action spaces to robotic manipulation. A popular approach is to do $Q$ learning with a dense pixel action space using a fully convolutional neural network (this is the FCN approach we describe and extend in Section 4.2) [18, 19, 20, 21]. Variations on this approach have been explored in [22, 23]. The FCN approach has been adapted to a variety of different manipulation tasks with different action primitives [24, 25, 26, 27, 28, 1, 29, 30, 31]. In this paper, we extend the work above by proposing new equivariant architectures for the spatial action space setting.

## 3 Problem Statement

We are interested in solving complex robotic manipulation problems such as the packing and construction problems shown in Fig 1. We focus on problems expressed in a spatial action space. This section identifies conditions under which the $Q$ function is SE(2)-invariant. The next section describes how these invariance properties translate into equivariance properties in the neural network.

Manipulation as an MDP in over a visual state space and a spatial action space: We assume that the manipulation problem is formulated as a Markov decision process (MDP): $\mathcal{M} = (S, A, T, R, \gamma)$. We focus on MDPs in visual state spaces and spatial action spaces [29, 20, 31]. The state space is factored into the state of the objects in the world, expressed as an $n$-channel $h \times w$ image $I \in S_{\text{world}} = \mathbb{R}^{n \times h \times w}$, and the state of the robot (including objects held by the robot) $s_{\text{rbt}} \in S_{\text{rbt}}$, expressed arbitrarily. The total state space is $S = S_{\text{world}} \times S_{\text{rbt}}$. The action space is expressed as a cross product of SE(2) (hence it is spatial) and a set of additional arbitrary action variables: $A = \text{SE}(2) \times A_{\text{arb}}$. The spatial component of action expresses where the robot hand is to move and the additional action variables express how it should move or what it should do. For example, in the pick/place domains shown in Fig 1, $A_{\text{arb}} = \{\text{PICK}, \text{PLACE}\}$, giving the agent the ability to move to a pose and close the fingers (pick) or move and open the fingers (place). We will sometimes decompose the spatial component of action $a_{\text{sp}} \in \text{SE}(2)$ into its translation and rotation components, $a_{\text{sp}} = (x, \theta)$. The goal of manipulation is to achieve a desired configuration of objects in the world, as expressed by a reward function $R : S \times A \to \mathbb{R}$.

Translation and Rotation in SE(2): We are interested in learning policies that are invariant to translation and rotation of the state and action. To do that, we define rotation and translation of state and action as follows. Let $g \in \text{SE}(2)$ be an arbitrary rotation and translation in the plane and let

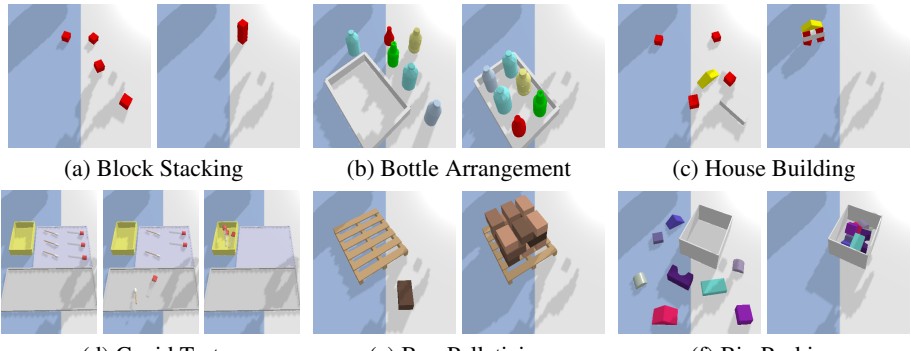

(a) Block Stacking      (b) Bottle Arrangement      (c) House Building

(d) Covid Test      (e) Box Palletizing      (f) Bin Packing

Figure 1: The experimental environments implemented in PyBullet [32]. The left image in each sub figure shows an initial state of the environment; the right image shows the goal state.

$s = (I, s_{\mathrm{rbt}}) \in S_{\mathrm{world}} \times S_{\mathrm{rbt}}$ be a state. $g$ operates on $s$ by rotating and translating the image $I$, but leaving $s_{\mathrm{rbt}}$ unchanged: $gs = (gI, s_{\mathrm{rbt}})$, where $gI$ denotes the image $I$ translated and rotated by $g$. For action $a = (a_{\mathrm{sp}}, a_{\mathrm{arb}})$, $g$ rotates and translates $a_{\mathrm{sp}}$ but not $a_{\mathrm{arb}}$: $ga = (ga_{\mathrm{sp}}, a_{\mathrm{arb}})$. Notice that both $S$ and $A$ are closed under $g \in \mathrm{SE}(2)$, i.e. that $\forall g \in \mathrm{SE}(2)$, $a \in A \implies ga \in A$ and $s \in S \implies gs \in S$.

Assumptions: We assume that the reward and transition dynamics of the system are invariant with respect to translation and rotation of state and action as defined above, and that the translation and rotation operations on state and action are invertible.

**Assumption 3.1** (Goal Invariance). *The manipulation objective is to achieve a desired configuration of objects in the world without regard to the position and orientation of the scene. That is, $R(s, a) = R(gs, ga)$ for all $g \in \mathrm{SE}(2)$.*

**Assumption 3.2** (Transition Invariance). *The outcome of robot actions is invariant to translations and rotations of both the scene and the action. Specifically, $T(s, a, s') = T(gs, ga, gs')$ for all $g \in \mathrm{SE}(2)$.*

**Assumption 3.3** (Invertibility). *Translations and rotations in state and action are invertible. That is, $\forall g \in \mathrm{SE}(2)$, $g^{-1}(gs) = s$ and $g^{-1}(ga) = a$.*

Assumptions 3.1 and 3.2 are satisfied in problem settings where the objective and the transition dynamics can be expressed intrinsically to the world without reference to an external coordinate frame imposed by the system designer. These assumptions are satisfied in many manipulation domains including all those shown in Fig 1. In House Building, for example, the reward and transition dynamics of the system are independent of the coordinate frame of the image or the action space. Assumption 3.3 is needed to guarantee the $Q$ function invariance described in the next section.

## 4 Approach

Assumptions 3.1, 3.2, and 3.3 imply that the optimal $Q$ function is invariant to translations and rotations in $\mathrm{SE}(2)$.

**Proposition 4.1.** *Given an MDP $\mathcal{M} = (S, A, T, R, \gamma)$ for which Assumptions 3.1, 3.2, and 3.3 are satisfied, the optimal $Q$ function is invariant to translation and rotation, i.e. $Q^*(s, a) = Q^*(gs, ga)$, for all $g \in \mathrm{SE}(2)$. (Proof in Appendix A.)*

Our key idea is to use the invariance property of Proposition 4.1 to structure $Q$ learning (and make it more sample efficient) by defining a neural network that is hard-wired to encode only invariant $Q$ functions. However, in order to accomplish this in the context of DQN, we must allow for the fact that state is an input to the neural network while action values are an *output*. This neural network is therefore a function $q : S \to \mathbb{R}^A$, where $\mathbb{R}^A$ denotes the space of functions $\{A \to \mathbb{R}\}$. The invariance property of Proposition 4.1 now becomes an *equivariance* property,

$$q(gs)(a) = q(s)(g^{-1}a), \tag{1}$$

where $q(s)(a)$ denotes the $Q$ value of action $a$ in state $s$. We implement this constraint using equivariant convolutional layers as described below.

### 4.1 Equivariant Convolutions

Equivariance over a finite group: In order to implement the equivariance constraint, it is standard in the literature to approximate $\mathrm{SE}(2)$ by a finite subgroup [4, 14]. Recall that the spatial component of an action is $a_{\mathrm{sp}} = (x, \theta) \in \mathrm{SE}(2)$. We constrain position to be a discrete pair of positive integers $x \in \{1 \ldots h\} \times \{1 \ldots w\} \subset \mathbb{Z}^2$, corresponding to a pixel in the input image $I$. We constrain orientation to be a member of a finite cyclic group $\theta \in C_u$, i.e. one of $u$ discrete orientations. For example, if $u = 8$, then $C_8 = \{0, \frac{\pi}{4}, \frac{2\pi}{4}, \frac{3\pi}{4}, \frac{4\pi}{4}, \frac{5\pi}{4}, \frac{6\pi}{4}, \frac{7\pi}{4}\}$. Our finite approximation of $a_{\mathrm{sp}} \in \mathrm{SE}(2)$ is $\hat{a}_{\mathrm{sp}}$ in the subgroup $\hat{\mathrm{SE}}(2)$ generated by translations $\mathbb{Z}^2$ and rotations $C_u$.

Input and output of an equivariant convolutional layer: A standard convolutional layer $h$ takes as input an $n$-channel feature map and produces an $m$-channel map as output, $h_{\mathrm{standard}} : \mathbb{R}^{n \times h \times w} \to \mathbb{R}^{m \times h \times w}$. We can construct an equivariant convolutional layer by adding an additional dimension to the feature map that encodes the values for each element of a group ($C_u$ in our case).[1] The equivariant mapping therefore becomes $h_{\mathrm{equiv}} : \mathbb{R}^{u \times n \times h \times w} \to \mathbb{R}^{u \times m \times h \times w}$ for all layers except the first. The first layer of the network generally takes a "flat" image as input: $h_{\mathrm{equiv}}^{\mathrm{in}} : \mathbb{R}^{1 \times n \times h \times w} \to \mathbb{R}^{u \times m \times h \times w}$.[2]

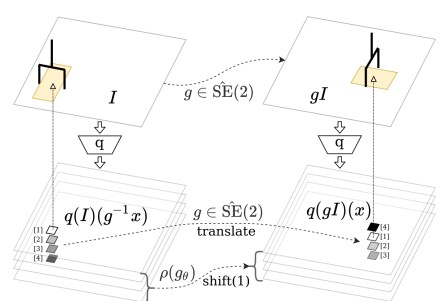

Figure 2: Illustration of $Q$-map equivariance when $C_u = C_4$. The output $Q$-map rotates and translates with the input image. The 4-vector at each pixel does a circular shift, i.e., the optimal rotation changes from 0 (the 1st element of $C_4$) to $\frac{\pi}{2}$ (the 2nd element of $C_4$)

Equivariance constraint: Let $h_i(I)(x)$ denote the output of convolutional layer $h$ at channel $i$ and pixel $x$ given input $I$. For an equivariant layer, $h_i(I)(x) \in \mathbb{R}^{C_u}$ describes feature values for each element of $C_u$. For an element $g \in \hat{\mathrm{SE}}(2)$, denote the rotational part by $g_\theta \in C_u$. If we identify functions $\mathbb{R}^{C_u}$ with vectors $\mathbb{R}^u$, then the group action of $g_\theta \in C_u$ on $\mathbb{R}^{C_u}$ becomes left multiplication by a permutation matrix $\rho(g_\theta)$ that performs a circular shift on the vector in $\mathbb{R}^u$. Then the group action of $\hat{\mathrm{SE}}(2)$ on a feature map $h(I) \in \mathbb{R}^{u \times m \times h \times w}$ can be expressed as $g(h_i(I))(x) = \rho(g_\theta)h_i(I)(g^{-1}x)$. The mapping $h$ is equivariant if and only if

$$h_i(gI)(x) = g(h_i(I))(x) = \rho(g_\theta)h_i(I)(g^{-1}x), \tag{2}$$

for each $i \in \{1 \ldots m\}$. This is illustrated in Fig 2. We can calculate the output feature map in the lower right corner by transforming the input by $g$ and then doing the convolution (left side of Eq. 2) or by doing the convolution first and then taking the value of $g^{-1}x$ and circular-shifting the output vector (right side of Eq. 2). In order to create a network that enforces the constraint of Eq. 1, we can simply stack equivariant convolutions layers that each satisfy Eq. 2.

Kernel constraint: The equivariance constraint of Eq. 2 can be implemented by strategically tying weights together in the convolutional kernel [4]. Since the standard convolutional kernel is already translation equivariant [13], we must only enforce rotational ($C_u$) equivariance [33]:

$$K(g_\theta y) = \rho_{\mathrm{out}}(g_\theta)K(y)\rho_{\mathrm{in}}(g_\theta)^{-1}, \tag{3}$$

where $\rho_{\mathrm{in}}(g_\theta)$ and $\rho_{\mathrm{out}}(g_\theta)$ are the permutation matrix of the group element $g_\theta$ (note that for the first layer, $K(y)$ will be a $1 \times u$ matrix, and $\rho_{\mathrm{in}}(g_\theta)$ will be 1). More details are in Appendix B.

### 4.2 Equivariant Fully Convolutional $Q$ Functions in $\mathrm{SE}(2)$

A baseline approach to encoding the $Q$ function over a spatial action space is to use a fully convolutional network (FCN) that stacks convolutional layers to produce an output $Q$ map with the same resolution as the input image. If we ignore the non-image state variables $s_{\mathrm{rbt}}$ and the non-spatial action variables $a_{\mathrm{arb}}$, then we have all the tools we need – we simply replace all convolutional layers with equivariant convolutions and the $Q$ network becomes fully equivariant.

Partial Equivariance: Unfortunately, in realistic robotics problems, $Q$ function is generally not equivariant with respect to all state and action variables. For example, the non-equivariant parts of state

---

[1]In the language of [14], this is a steerable convolution between regular representations of $C_u$.

[2]This is a steerable convolution between the trivial representation and regular representation of $C_u$.

and action in Section 3 are $s_{\text{rbt}}$ and $a_{\text{arb}}$. We encode $a_{\text{arb}}$ by simply having a separate output head for each. However, to encode $s_{\text{rbt}}$, we need a mechanism for inserting the non-equivariant information into the neural network model without "breaking" the equivariance property. We explored two approaches: the lift expansion ap-

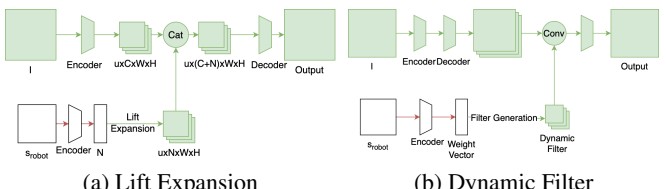

(a) Lift Expansion      (b) Dynamic Filter

Figure 3: Illustration of approaches to partial equivariance. Green blocks are equivariant and white blocks are not.

proach and the dynamic filter approach. In the lift expansion, we tile the non-equivariant information across the equivariant dimensions of the feature map as additional channels (Fig 3a). In the dynamic filter approach [34], the non-equivariant data is passed through a separate pathway that outputs the weights of an equivariant kernel that is convolved into the main equivariant backbone. We constrain this filter to be equivariant by enforcing the kernel constraint of Eq. 3 (Fig 3b). We empirically find that both methods have similar performance (Appendix G.2). In the remainder of this paper, we use the dynamic filter approach because it is more memory efficient.

Encoding Gripper Symmetry Using Quotient Groups: Another symmetry that we want to leverage is the bilateral symmetry of the gripper. The outcome of a pick action performed using a two-finger gripper in orientation $\theta$ is the same as for the gripper in orientation $\theta + k\pi$ for any integer $k$. Similarly, it is often valid to assume that the outcome of place actions is invariant [3]. We model this invariance using the quotient group $C_u/C_2$. The $C_2 = \{0, \pi\}$ action equates rotations which differ by multiples of $\pi$ in $C_u/C_2$. The steerable layer defined under the quotient group is applied with the same constraint as in Eq. 3, except that the output space will be in $C_u/C_2$.

Experimental Domains: We evaluate the equivariant FCN approach in the Block Stacking and Bottle Arrangement tasks shown in Fig 1. Both environments have sparse rewards (+1 at goal and 0 otherwise). The world state is encoded by a 1-channel heightmap $I \in \mathbb{R}^{1 \times h \times w}$ and robot state is encoded by an image patch $H$ that describes the contents of the robotic hand. The non-spatial action variable $a_{\text{arb}} \in \{\text{PICK}, \text{PLACE}\}$ is selected by the gripper state, i.e., $a_{\text{arb}} = \text{PLACE}$ if the gripper is holding an object, and PICK otherwise. The equivariant layers of the FCN are defined over group $C_{12}$ where the output is with respect to the quotient group $C_{12}/C_2$ to encode the gripper symmetry. See Appendix C and D for detail on the experimental domains and the FCN architecture respectively.

Experimental Comparison With Baselines: We evaluate against the following baselines: 1) Conventional FCN: FCN with 1-channel input and 6-channel output where each output channel corresponds to a $Q$ map for one rotation in the action space (similar to Satish et al. [19] but without the $z$ dimension). 2) RAD [7] FCN: same architecture as 1), while at each training step, we augment each transition in the minibatch with a rotation randomly sampled from $C_{12}$. 3) DrQ [8] FCN: same architecture as 1), while at each training step, the $Q$ targets and $Q$ outputs are calculated

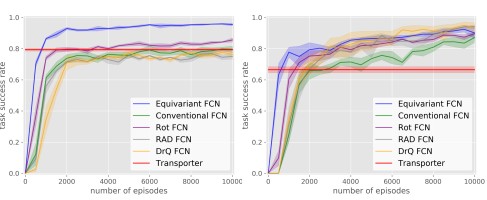

(a) Block Stacking      (b) Bottle Arrangement

Figure 4: Comparison of Equivariant FCN (blue) with baselines. Results averaged over four runs. Shading denotes standard error.

by averaging over multiple augmented versions of the sampled transitions. Random rotations sampled from $C_{12}$ are used for the augmentation. 4) Rot FCN: FCN with 1-channel input and 1-channel output, the rotation is encoded by rotating the input and output for each $\theta$ [21]. 5) Transporter Network [1], an FCN-based architecture with the last layer being a dynamic kernel generated by a separate FCN with an input of an image crop at the pick location. Baseline 2) and 3) are data augmentation methods that aim to learn the symmetry encoded in our equivariant network using rotational data augmentation sampled from the same symmetry group ($C_{12}$) as used by our equivariant model. All baselines have the same action space as the proposal. More detail on the baselines is in Appendix E.1. All methods except the Transporter Network use SDQfD, an approach to imitation learning in spatial action spaces that combines a TD loss term with penalties on non-expert actions [29]. (Transporter Network is a behavior cloning method.) Table 1 shows the number of demonstration steps. Those expert transitions are augmented by 9 random

---

[3] Strictly speaking, this is true only when the grasped object is also symmetric.

| | Block Stacking | Bottle Arrangement | House Building | Box Palletizing | Covid Test | Bin Packing |
|---|---|---|---|---|---|---|
| expert steps | 50 | 240 | 200 | 1000 | 2000 | 2000 |
| equivalent episodes | 8 | 20 | 20 | 28 | 111 | 125 |

Table 1: The number of expert steps and the (approximate) equivalent number of episodes.

$SE(2)$ transformations. See Appendix F.2 for more parameter detail. Fig 4 shows the results. Our equivariant FCN outperforms all baselines in the block stacking task. Notice that in the Bottle Arrangement task, the equivariant network learns faster than the baselines but converges to a similar level as RAD and DrQ. This is because the domain itself is already partially rotationally equivariant because the bottles are cylindrical and therefore our network has less of an advantage.

### 4.3 Equivariant Augmented State $Q$ Functions in $SE(2)$

The FCN approach does not scale well to challenging manipulation problems. Therefore, we design an equivariant version of the augmented state representation (ASR) method of [29], which has been shown to be faster and have better performance. The ASR method transforms the original MDP with a high dimensional action space into a new MDP with an augmented state space but a lower dimensional action space. Instead of encoding the value of all dimensions of action in a single neural network, this model encodes the value of different factorized parts of the action space such as position and orientation using separate neural networks conditioned on prior action choices.

ASR in $SE(2)$: We explain the ASR method in the example setting of the $SE(2)$ action space. See Fig 5 for an illustration. As before, actions $\hat{a}_{sp} = (x, \theta)$ are elements of the space $\hat{SE}(2)$, the finite approximation of $SE(2)$ as in Section 4.1. However, the $Q$ function is now computed using two separate functions, the position function $Q_1(s, x) = \max_\theta Q(s, (x, \theta))$ and the orientation function $Q_2((s, x), \theta) = Q(s, (x, \theta))$. $Q_1$ is encoded using a fully convolutional network $q_1 : \mathbb{R}^{n \times h \times w} \to \mathbb{R}^{1 \times h \times w}$ that takes an $n$-channel image $I$ as input and produces a 1-channel $Q$ map that describes $Q_1(s, x)$ for all $x$. We evaluate $Q_2$ on the "augmented state" $(s, x)$ which contains the state $s$ and the chosen $x$. The augmented state

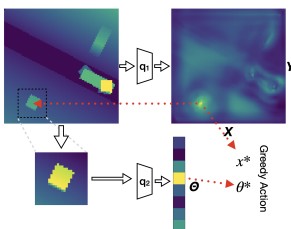

Figure 5: Illustration of the ASR approach in $SE(2)$.

is encoded using the image patch $P = \text{CROP}(I, x) \in \mathbb{R}^{n \times h' \times w'}$ cropped from $I$ and centered at $x$. We model $Q_2$ using the network $q_2 : \mathbb{R}^{n \times h' \times w'} \to \mathbb{R}^u$ that takes input $P$ and outputs $Q_2((s, x), \theta)$ for all $u$ different orientation $\theta$. These two networks are used together for both action selection and evaluation of target values during learning. We evaluate $x^* = \arg\max_x q_1(I)$, calculate $P = \text{CROP}(I, x^*)$, and then evaluate $\theta^* = \arg\max_\theta q_2(P)$ and $Q^* = \max_\theta q_2(P)$. Note that $Q$ maps produced by $q_1$ and $q_2$ are of size $u + hw$, significantly smaller than the $Q$ map in the FCN approach which is size $uhw$. Essentially the ASR method takes advantage of the fact that the optimal $\theta$ depends only on the local patch $P$ given an optimal position $x$.

Equivariant architecture for ASR in $SE(2)$: We decompose the $SE(2)$ equivariance property of Eq. 1 into two equivariance properties for $q_1$ and $q_2$, respectively: $q_1(gI)(x) = q_1(I)(g^{-1}x)$ where $g \in \hat{SE}(2)$, and $q_2(g_\theta P) = \rho(g_\theta)q_2(P)$ where $g_\theta \in C_u$. The equivariance property of $q_1$ is similar to that of Eq. 2 except that the output of $q_1$ has only one channel which is invariant to rotations (since it is a maximum over all rotations). This means we can rewrite the $q_1$ equivariance property as $q_1(gI) = gq_1(I)$, where $g$ on the RHS of this equation translates and rotates the output $Q$ map. In practice, we obtained the best performance for $q_1$ by enforcing equivari-

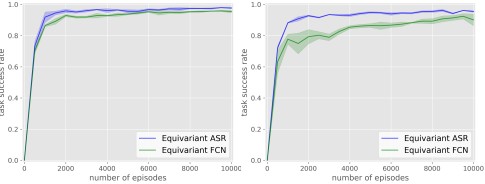

(a) Block Stacking    (b) Bottle Arrangement

Figure 6: Comparison between Equivariant ASR (blue) and Equivariant FCN (green). Results averaged over four runs. Shading denotes standard error.

ance to the Dihedral group $D_4$, which is generated by 90 degree rotation and reflections over the coordinate axis. For $q_2$, we used an equivariant feature map that outputs a single $u$-dimensional vector of $Q$ values corresponding to the finite cyclic group $C_u$ used. (We use $C_{12}/C_2$ and $C_{32}/C_2$ in our experiments below). We handle the partial equivariance using the same strategies as earlier. Appendix D describes the model details.

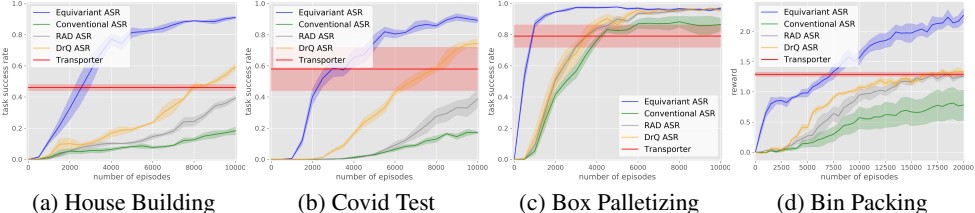

|          (a) House Building          |          (b) Covid Test          |          (c) Box Palletizing          |          (d) Bin Packing          |

Figure 7: Comparison of Equivariant ASR (blue) with baselines. Results averaged over four runs. Shading denotes standard error.

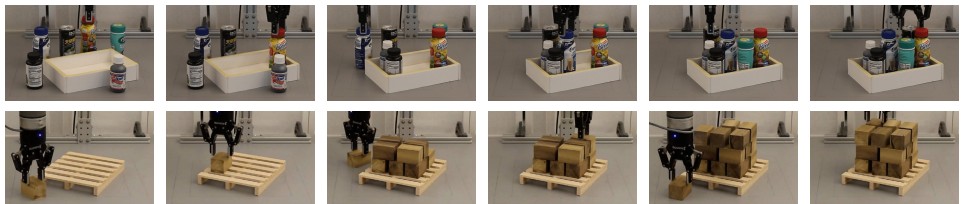

Figure 8: Top row: the robot finishing the Bottle Arrangement task. Bottom row: the robot finishing the Box Palletizing task. Full episodes are in Appendix I.

**Experimental Comparison with Equivariant FCN:** Fig 6 shows a comparison between equivariant ASR (this section) and equivariant FCN (Section 4.2) for the Block Stacking and Bottle Arrangement tasks. The network $q_2$ is defined using $C_{12}$ and its quotient group $C_{12}/C_2$ to match Section 4.2. The ASR method surpasses the FCN method in both tasks.

**More Challenging Experimental Domains:** The equivariant ASR method is able to solve more challenging manipulation tasks than equivariant FCN can. In particular, we could not run the FCN with as large a rotation space because it requires more GPU memory. We evaluate on the following four additional domains: House Building, Covid Test, Box Palletizing (introduced in [1]), and Bin Packing (Fig 1(c-f)). All domains except Bin Packing have sparse rewards. In Bin Packing, the agent obtains a positive reward inversely proportional to the highest point in the pile after packing all objects. See Appendix C for more details about the environments. We now define $q_2$ using the group $C_{32}$ and its quotient group $C_{32}/C_2$, i.e., we now encode 16 orientations ranging from 0 to $\pi$. As in Section 4.2, we use the SDQfD loss term to incorporate expert demonstrations (except for Transporter Net which uses standard behavior cloning exclusively). The number of expert transitions provided is shown in Table 1. 9 random SE(2) augmentations are applied to the expert transitions.

**Experimental Comparison with Non-Equivariant Baselines:** We compare equivariant ASR against the following non-equivariant baselines: 1) Conventional ASR: ASR in SE(2) with conventional CNNs rather than equivariant layers. 2) RAD [7] ASR: same architecture as (1) but each minibatch is augmented with a random rotation. 3): DrQ [8] ASR: same architecture as (1) but each $Q$ target and $Q$ estimate are calculated by averaging over several augmented versions of the sampled transition. The augmentation in (2) and (3) is by random rotations sampled from $C_{32}$, the same group used in the equivariant model. 4) the Transporter Network [1]. See Appendix E.2 for the network architecture for the baselines. The results in Fig 7 show that equivariant ASR outperforms the other methods on all tasks, followed by DrQ, RAD, and Transporter Net, followed by Conventional ASR.

**Robot Experiment:** We evaluate the trained equivariant ASR models for Bottle Arrangement, House Building, and Box Palletizing on a Universal Robots UR5 arm equipped with a Robotiq 2F-85 gripper. The observation is provided by an Occipital Structure sensor mounted on top of the workspace. Table 2 shows the results. In Bottle Arrangement, the robot shows a 90% success rate. In one of the two failures, the arrangement is not compact enough, leaving no enough space left for the last bottle. In the other failure, the robot arranges the bottles outside of the tray. In the House Building task, the robot succeeds in all 20 episodes. In the Box Palletizing task, the robot demonstrates a 95% success

| Environment       | SR           |
|-------------------|--------------|
| Bottle Arrangement | 90%(18/20)  |
| House Building    | 100%(20/20) |
| Box Palletizing   | 95%(19/20)  |

Table 2: Robot experiment result

rate. In the failure, the robot correctly stacks 16 of 18 boxes, but the 17th box's placement position is offset slightly from the the rest of the stack and there is no room to place the last box. The same problem happens in another successful episode, where the fingers squeeze the boxes and make room for the last box. Fig 8 shows two example episodes in the robot experiment.

## 4.4 Equivariant Augmented State $Q$ Functions in $\mathrm{SE}(3)$

A strength of the ASR method is that it can be extended into $\mathrm{SE}(3)$ by adding networks similar to $q_2$ that encode $Q$ values for additional dimensions of the action space [29]. Specifically, we add three networks to the $\mathrm{SE}(2)$-equivariant architecture described in Section 4.3, $q_3$, $q_4$, and $q_5$ encoding $Q$ values for $Z$ (height above the plane), and angles $\Phi$ (rotation in XZ plane) and $\Psi$ (rotation in YZ plane) dimensions of $\mathrm{SE}(3)$. Each of these networks takes as input a stack of 3 orthographic projections of a point cloud along the coordinate planes. The point cloud is re-centered and rotated to encode the partial $\mathrm{SE}(3)$ actions. (see [29] for details).

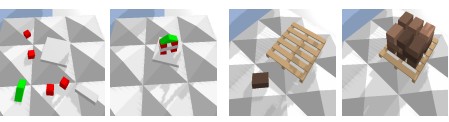

(a) House Building     (b) Box Palletizing

Figure 9: The 6DOF experimental domains.

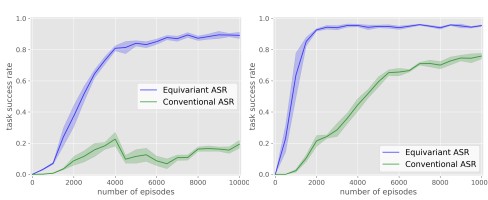

(a) House Building     (b) Box Palletizing 18

Figure 10: Comparison of the equivariant network with the baseline in 6DOF tasks. Results averaged over four runs. Shading denotes standard error.

Unfortunately, we cannot easily make these additional networks equivariant using the same methods we have proposed so far because they encode variation outside of $\mathrm{SE}(2)$. Instead, we create an encoding that is approximately equivariant by explicitly transforming the input to these networks in a way that corresponds to a set of candidate robot positions or orientations (called a "deictic" encoding in [22]). We will describe this idea using $q_2$ as an example. Define $q_2'(P) \in \mathbb{R}$ to output the single $Q$ value of the rotation encoded in the input image $P$. Then $q_2$ can be defined as a vector-valued function: $\hat{q}_2(P) = (q_2'(g_1^{-1}P), \ldots, q_2'(g_u^{-1}P))$, where $g_1, \ldots, g_u \in C_u$. $\hat{q}_2$ is approximately equivariant because everything learned by $q_2'$ is automatically replicated in all orientations. We design deictic $q_3$, $q_4$, and $q_5$ similarly by selecting a finite subset of $\{g_1, \ldots, g_K\} \subset \mathrm{SE}(3)$ corresponding to the dimension of the action space encoded by each $q_i$. $q_i$ can then be defined by evaluating a network $q_i'$ over input $P$ transformed by $(g_i)_{i=1}^K$. For $q_3$, we evaluate over 36 translations $g_k(z) = z + k(0.18/36) + 0.02$ where $0 \leq k \leq 35$. For $q_4$ and $q_5$, we use rotations $g_k \in \{-\frac{\pi}{8}, -\frac{\pi}{12}, -\frac{\pi}{24}, 0, \frac{\pi}{24}, \frac{\pi}{12}, \frac{\pi}{8}\}$. Note we use $q_2$ for explanation, while our model uses equivariant $q_1$, $q_2$ and deictic $q_3$-$q_5$. For an ablated version using deictic $q_2$, see Appendix G.4.3 and G.4.4. Comparison to Non-Equivariant Approaches: We evaluate ASR in $\mathrm{SE}(3)$ in the House Building and Box Palletizing domains. We modified those environments so that objects are presented randomly on a bumpy surface with a maximum out of plane orientation of 15 degrees (Fig 9). In order to succeed, the agent must correctly perform pick and place actions with the needed height and out of plane orientation. We evaluated the Equivariant ASR in comparison with a baseline Conventional ASR (same as [29]). Both methods use SDQfD with 2000 expert demonstration steps. The results are shown in Fig 10. Our proposed approach outperforms the baseline by a significant margin.

## 5 Discussion

In this paper, we show that equivariant neural network architectures can be used to improve $Q$ learning in spatial action spaces. We propose multiple approaches and model architectures that can be used to accomplish this and demonstrate improved sample efficiency and performance on several robotic manipulation applications both in simulation and on a physical system. This work has several limitations and directions for future research. First, our methods apply directly only to problems in spatial action spaces. While many robotics problems can be expressed this way, it would clearly be useful to develop equivariant models for policy learning that can be used in other settings. Second, although we extend our ASR approach from $\mathrm{SE}(2)$ to $\mathrm{SE}(3)$ in the last section of this paper, this solution is not fully equivariant in $\mathrm{SE}(3)$ and it may be possible to do better by exploiting methods that are directly equivariant in $\mathrm{SE}(3)$.

**Acknowledgments**

This work is supported in part by NSF 1724257, NSF 1724191, NSF 1763878, NSF 1750649, and NASA 80NSSC19K1474. R. Walters is supported by a Postdoctoral Fellowship from the Roux Institute and NSF grants 2107256 and 2134178.

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
