# OpenReview forum: "Equivariant $Q$ Learning in Spatial Action Spaces"
_robot-learning.org/CoRL/2021/Conference — CoRL2021 Poster_

### Official Review · Reviewer_BRhk · 2021-07-18

**Originality:** Good
**Technical Quality:** Very Good
**Clarity Of Presentation:** Very Good
**Impact:** 4

**Recommendation:**

Weak Accept: I recommend accepting the paper, but will not argue for my recommendation if the majority of other reviewers have a different opinion.

**Summary:**

The paper aims at exploiting equivariances under SE(2) present in robotics manipulation tasks with spatial state and action spaces in order to improve sample efficiency of image-based deep Q-learning.
In particular, the paper proposes to exploit the equivariance of the Q-function wrt. simultaneous symmetry transformations in the state and action spaces. For that purpose the authors introduce two new architectures for the network parametrizing the Q-function, which are equivariant wrt SE(2) transformations of (spatial components of) the state and action space.
The authors compare the performance of their proposed architectures to several baselines, which either do not take into account equivariances or try to learn these via data augmentation, in several robotics manipulation tasks in simulation. Furthermore, the authors demonstrate the applicabilty of their approach in a real world setup.

**Issues:**

- line 29: The authors state that other methods cited in previous lines improve generalization but "generally make policy learning harder". I think this statement is too strong and not properly justified.

- experiments:
-- I wonder how the angles for the outpur of basesline 1 (line 216) are chosen. Do they have the same granularity and respect the gripper symmetry as the equivariant method (corresponding to a choice [0, 30, 60, 90, 120, 150])? The authors should state that. Also, if the authors chose a coarsere granularity (i.e. [0, 60, 120, 180, 240, 320]) I believe the comparison is not fair.  Similarly I wonder, why the authors chose to rotate the input of the baseline 2) only in terms of 90 degree, leading to a much coarser angel granularity. I believe this comparison is not really fair.
-- The authors do not discuss the runtime of their approach. How much does the runtime suffer from the sequential approach the authors propose in section 4.3?
--  I think, that the assumed SE(2) equivariance may be slightly violated in the considered experiments due to the fact that the robot is fixed to the workspace at some point. Some state-action pairs might therefore be more difficult to attain for the robot than an equivariant pair. Did the authors observe such effects?
-- section 4.4: The architecture and approximation mentioned in lines 339ff is not sufficiently discussed. The reader has no chance to understand it without the appendix. I think that such important "details" should be part of the main paper and not be transferred to the appendix, otherwise a page limit would become meaningless.


**Reviewer Expertise:**

Very good: Comprehensive knowledge of the area

**Strengths And Weaknesses:**

Strenghts:
The paper is well written and mathematically sound.
It clearly outlines the pre-conditions under which the considered equivariance properties apply.
It proposes two novel architectures that directly encode the SE(2) equivariance into the structure of the network of the Q-function.
The experiments  are well chosen and results are properly documented and discussed.

Weaknesses:
see detailed issues below

**Summary Of Recommendation:**

The paper discusses an the topic of equivariances, which is very important for robotics research and applications. The proposed approach to encode the equivariance directly into the network of the Q-function is promising and presumably easy to implement and might therefore have an impact on robotics research.
The paper is well written, mathematically sound and the experiments are well chosen and properly discussed.
There are some minor issues, which the authors should address. Inspite of these I would recommend the paper for acceptance to CORL, since the positive aspects clearly outweigh the negatives.

---

> ### Author Response · Authors · 2021-08-20
> **Author Reponse to Reviewer BRhk**
>
> The authors thank the reviewer for the careful review. We are glad that the reviewer acknowledged that our paper is mathematically sound. Please see our comments below:
>
> The reviewer raised a concern that the claim in line 29 is too strong. Thanks for pointing this out. We revised this claim to "they require the neural network to learn translational and rotational invariance from the augmented data." (line 27 in the revision)
>
> The reviewer raised a question about how the angles of the output of baseline 1 were chosen. Please note that for each experiment, all methods have the same action granularity. We added a sentence in line 228 in the revision to clarify this.
>
> The reviewer further raised a concern about the fairness of the comparison to the soft equivariant baseline. We replaced the soft equivariant baseline with two standardized image augmentation baselines, RAD [1] and DrQ [2], as detailed in our AC comment.
>
> The reviewer asked if the runtime suffers from the sequential approach in section 4.3. For the same choice of rotation group, the ASR method of sect 4.3 is actually faster than the FCN approach because each of the sequential networks is smaller than the FCN network. We included a runtime analysis in Appendix H in the revision.
>
> The reviewer raised a concern about the possible violation of SE(2) equivariance due to the robot arm's workspace limitation. While it is true that our system may be non-equivariant at the edges of the workspace, we constrain the workspace such that this problem does not arise.
>
> The reviewer raised a concern that the method of section 4.4 is hard to understand without referring to the appendix. We included more detail in section 4.4 in the revision.
>
> [1] Laskin, Michael, et al. "Reinforcement learning with augmented data." arXiv preprint arXiv:2004.14990 (2020).
> [2] Kostrikov, Ilya, Denis Yarats, and Rob Fergus. "Image augmentation is all you need: Regularizing deep reinforcement learning from pixels." arXiv preprint arXiv:2004.13649 (2020).

---

> > ### Comment · Reviewer_BRhk · 2021-08-26
> > **Reply to auhtors**
> >
> > Thank you for your reply and the clarifications.
> > As for now, I will wait until for the final revision of the paper before re-assessing.

---

### Official Review · Reviewer_GkB1 · 2021-07-24

**Originality:** Good
**Technical Quality:** Good
**Clarity Of Presentation:** Good
**Impact:** 4

**Recommendation:**

Strong Accept: I recommend accepting the paper and will argue for my recommendation even if other reviewers hold a different opinion.

**Summary:**

This paper presents an architecture for learning rotation and translation equivariant policies in order to learn object manipulation policies more efficiently. The method also provides a channel for non-equivariant information. The experiments are on 6 simulated pick-and-place tasks with real world evaluations.

**Issues:**

See above.

**Reviewer Expertise:**

Very good: Comprehensive knowledge of the area

**Strengths And Weaknesses:**

Strengths:
- The topic is relevant to CoRL. Increasing learning efficiency is an important problem in robot learning. Structural biases such as rotational and translational equivariance are sensible and applicable to many manipulation problems.
- The experimental results show the method is able to perform well on a variety of pick and place tasks.
- The soft-equivariant baseline is a nice comparison of data enforced equivariance vs model enforced. Does " augmented three fold for rotations every 90 degrees" match the number of rotations in the model? If not, then having an experiment where they do match would be the best comparison.

Clarity:
- The method is very confusing to read. The architecture is based of off ASR [29], but ASR is barely explained. It would be easier to understand if the equivariant aspect were first explained for a normal FCN policy, and then the ASR was added. My understanding is that with ASR, each dimension of the action space is predicted sequentially. From the video, the policy looks like a simple FCN applied to each rotation of the image, but this is not what I understood from reading the paper. I don't understand how this architecture differs from the one in Form2Fit, which outputs embeddings for each of 20 rotations (I do understand that algorithmically, this paper outputs Q values and Form2Fit outputs embeddings, is that the only difference?).
- The theoretical backing for the method is overwrought for what seems to be in the a fairly simple method, consider cleaning it up to keep only what is necessary.

Experiments:
- To disentangle the contributions of Q learning vs this approach to equivariance, the comparison to transporter could be made more fair by either evaluating a BC version of the proposed model or a Q-learning method of the transporter networks.

Question:
- Is the model always applied to images where the gripper is not visible?

**Summary Of Recommendation:**

The paper's clarity needs to be improved. It is difficult to figure out exactly what the method is and how it differs from prior work. This also makes the novelty of the method difficult to determine. I find the general direction of the work to be impactful, although the method seems limited to non-dynamic pick and place tasks.


-----------

Update after author response and improved paper
I find the paper clarity improved and the new experiments convincing. I am in favor of accepting this paper.

---

> ### Author Response · Authors · 2021-08-20
> **Author Response to Reviewer GkB1**
>
> The authors thank the reviewer for the careful review. In response to the reviewer's concern about the "soft equivariant" baseline, we replaced the "soft equivariant" baseline with RAD [1] and DrQ [2] implemented to make rotational augmentations using the same angular discretization as in our proposed methods, as detailed in our AC comment.
>
> The reviewer raised a concern about the clarity of our write-up. We have updated section 4 to clarify these issues. We include a summary here as well.
>
> - The review asked that we describe equivariance for a "normal FCN policy". Please note that this is exactly what we do in sections 4.1 and 4.2. We only introduce ASR in section 4.3.
>
> - The reviewer thought that "From the video, the policy looks like a simple FCN applied to each rotation of the image". In the video, we explain that the optimal Q function has a given equivariance property: if the input were to be rotated, the output would change in a predictable manner. By encoding this mathematical property into the network using equivariant neural networks (which is the contribution of this work), we actually make it unnecessary to pass rotated versions of the image to the network. In fact, equivariant neural networks can automatically generalize to rotated images.
>
> - The main distinction between our equivariant FCN approach (Section 4.1 and 4.2) and "a simple FCN applied to each rotation of the image" (Rot FCN in short) is that equivariant FCN does not require passing all rotated images to the same network to encode the rotation, which dramatically increases the GPU consumption. Equivariant FCN instead enforces the rotational generalization through weight tieing. We also show experimentally that the equivariant FCN outperforms the Rot FCN method in Figure 4. In addition, the equivariant network can be applied to the ASR approach (Section 4.3) and further extends to an SE(3) action space (Section 4.4).
>
> - The reviewer asked, "how this architecture differs from the one in Form2Fit, which outputs embeddings for each of 20 rotations". Form2Fit encodes rotations in the same way as Rot FCN (i.e., passing all rotated versions of the same image to the same network), while our model incorporates rotational equivariance into the network model in a way that Form2Fit does not. It would be possible to create an equivariant version of Form2Fit, which could be a direction of future work.
>
> The reviewer further raised a concern that "the theoretical backing for the method is overwrought for what seems to be a fairly simple method". We felt a certain amount of theoretical backing was required for our method as equivariant methods are non-trivial as they involve layers mathematically constrained to follow certain symmetries. In addition, our network accommodates both non-equivariant and equivariant inputs in a novel "partially equivariant" architecture.
>
> The reviewer suggested "To disentangle the contributions of Q learning vs this approach to equivariance, the comparison to transporter could be made more fair by either evaluating a BC version of the proposed model or a Q-learning method of the transporter networks". We did find that a BC version of our equivariant network outperforms the transporter network (Appendix G.3). We also have tried to implement a Q learning version of the transporter network, but it did not work as well as the original BC version.
>
> The reviewer also questioned if the model is always applied to images where the gripper is not visible. Our experiments in this paper only apply the method to situations where the gripper is not visible. However, our subsequent unpublished work shows that this is not essential to the approach.
>
> [1] Laskin, Michael, et al. "Reinforcement learning with augmented data." arXiv preprint arXiv:2004.14990 (2020).
> [2] Kostrikov, Ilya, Denis Yarats, and Rob Fergus. "Image augmentation is all you need: Regularizing deep reinforcement learning from pixels." arXiv preprint arXiv:2004.13649 (2020).

---

> > ### Comment · Reviewer_GkB1 · 2021-09-03
> > **Update to response**
> >
> > Thank you for updating the paper, I find it clearer and the new experiments convincing. I like the work and hope that the environments and code will be released publicly.

---

### Official Review · Reviewer_HHEb · 2021-07-26

**Originality:** Good
**Technical Quality:** Good
**Clarity Of Presentation:** Good
**Impact:** 3

**Recommendation:**

Weak Accept: I recommend accepting the paper, but will not argue for my recommendation if the majority of other reviewers have a different opinion.

**Summary:**

This paper proposes to bake in the equivariance constraints in the state-action space in deep Q-learning methods for robotic manipulation problems. The method is to utilize equivariant convolutions in addition to other symmetries (such as gripper symmetries) and additional state fed into the network. These new q-function architectures are then evaluated on a number of problems and are shown to be better.

**Issues:**

Atleast the data augmentation baseline must be added.

**Reviewer Expertise:**

Fair: Some knowledge of the area

**Strengths And Weaknesses:**

Strengths:
1. The paper proposes to actually utilize symmetries in the Q-function training, which is interesting since it can be done by just using a different architecture.
2. The empirical sample-efficiency results seem quite better compared to the baselines on a number of domains.

Weaknesses:
1. The paper does not compare to some obvious baselines such as enforcing these symmetries via data augmentation. While this is discussed in related work, a comparison on at least some domains is needed.
2. I would suggest reorganizing the paper to have a clearly defined method section with a summarized table of various options and then experiments.

**Summary Of Recommendation:**

The idea proposed in the paper is interesting, but I am not sure of the baselines (e.g., data augmentation) and how important the method will be in the long run -- is it likely that we will be able to just use simpler techniques without having to modify the architectures for Q-functions and obtain similar sample efficiency?

---

> ### Author Response · Authors · 2021-08-20
> **Author Response to Reviewer HHEb**
>
> The authors thank the reviewer for the careful review. In response to the reviewer’s concern about the missing comparison to some augmentation baselines. Please note that our “soft equivariant” baseline is a data augmentation approach that learns the rotational symmetry using data augmentation. However, since there are some concerns about the fairness of the “soft equivariant” baseline, as is pointed out by the AC, Reviewer GkB1, and Reviewer BRhk, we replaced the “soft equivariant” baseline with two more standardized baselines, RAD [1] and DrQ [2], as detailed in our AC comment.
>
> [1] Laskin, Michael, et al. "Reinforcement learning with augmented data." arXiv preprint arXiv:2004.14990 (2020).
> [2] Kostrikov, Ilya, Denis Yarats, and Rob Fergus. "Image augmentation is all you need: Regularizing deep reinforcement learning from pixels." arXiv preprint arXiv:2004.13649 (2020).

---

### Official Review · Reviewer_3ab5 · 2021-07-26

**Originality:** Very Good
**Technical Quality:** Excellent
**Clarity Of Presentation:** Good
**Impact:** 4

**Recommendation:**

Weak Accept: I recommend accepting the paper, but will not argue for my recommendation if the majority of other reviewers have a different opinion.

**Summary:**

The paper studies the relevance and benefits of using SE(2) equivariant models in Q-learning in robotic problems. The paper identifies the conditions under which the Q function is SE(2) equivariant (Proposition 4.1). The paper, via experimental evaluation on various robotic manipulation problems, demonstrates the sample efficiency of the equivariant models over other models considered in the literature.

**Issues:**

Following are some of the issues, but a major re-drafting of the paper write-up would help the paper make its case even better.

- Figure 3 should be placed close to page 2 where it is first referred to.

- In Sec 3, the paper states “learning policies that are invariant to translation and rotation.” This is misleading and confusing. The policies need to be invariant to translation and rotation to state space is perhaps what they mean.

- The abstract, and the write-up in general, are very verbose. Examples:
** “we identify the conditions under which the optimal Q function is SE(2) equivariant, and equivariant Q learning is therefore applicable.”
** Sentences 3-5 in the abstract.

It would have been good to briefly describe sample efficiency earlier in the Introduction as it is referred to so many times in the paper.


**Reviewer Expertise:**

Very good: Comprehensive knowledge of the area

**Strengths And Weaknesses:**

Strengths: The paper studies an important problem. The study of equivariant/invariant models for Q-learning and imposing structure gleaned from the problem, is a great direction to pursue. This paper presents a good step in that direction. Prior works have not used equivariant architectures to model Q functions and thus this paper is novel and makes a worthy contribution. The experimental results are thorough and convincing.

Weaknesses: The paper could have been better written.

**Summary Of Recommendation:**

The paper studies the relevance and benefits of using SE(2) equivariant models in Q-learning in robotic problems. The paper identifies the conditions under which the Q function is SE(2) equivariant (Proposition 4.1). The paper, via experimental evaluation on various robotic manipulation problems, demonstrates the sample efficiency of the equivariant models over others considered in the literature.
Prior works have not used equivariant architectures to model Q functions and thus this paper is novel and makes a worthy contribution. The paper is not very well written.

---

> ### Author Response · Authors · 2021-08-20
> **Author Response to Reviewer 3ab5**
>
> The authors thank the reviewer for the careful review. In response to the reviewer’s concern about the clarity of the write-up, we appreciate the editing suggestions. We revised the write-up in the revision, including addressing all of the three suggestions that the reviewer made.

---

### Author Response · Authors · 2021-08-27
**Summary of Revision**

The authors thank all reviewers for the careful review. We have made several changes in the revision based on the reviewer’s suggestions to make our paper stronger. We have uploaded the revision with the major changes labeled in blue. Here is a summary of the changes:

- Added the comparison against two strong augmentation baselines, RAD [1] and DrQ [2], in both Section 4.2 and Section 4.3. RAD and DrQ are implemented to make rotational augmentations using the same angular discretization as in our proposed methods.

- Revised Section 4.3 to improve the description of ASR in our method.

- Revised Section 4.4 to include more details.

- Added a definition of “sample efficiency” in Section 1.

- Revised the claim of “data augmentation methods make learning harder” in Section 1.

- Revised the write-up in general to improve the clarity.

- Added a runtime analysis in Appendix H.

- Reran the simulation box palletizing experiment using the same environment as in the robot experiment to improve the consistency.

[1] Laskin, Michael, et al. "Reinforcement learning with augmented data." arXiv preprint arXiv:2004.14990 (2020).
[2] Kostrikov, Ilya, Denis Yarats, and Rob Fergus. "Image augmentation is all you need: Regularizing deep reinforcement learning from pixels." arXiv preprint arXiv:2004.13649 (2020).

---

### Meta-Review · Area_Chair_afcK · 2021-08-13

**Recommendation:** Accept (Poster)
**Confidence:** 4

**Metareview:**

The authors propose in this paper a Q function neural architecture that is invariant to rotations. The consensus among reviewers is the following.

Pros:
- The problem is important and applicable to many tasks.
- The robotics task is well chosen and results are encouraging.

Cons:
- Concerns regarding the fairness of the experimental setup, especially the data augmentation baselines (R-H (Reviewer HHEb), R-G, R-B).
- The Q function architecture beyond the equivarient convolutions not well explained (R-G).

The authors are encouraged to actively engage with the reviewers and revise their submission if they see fit. Concerns regarding the fairness of the comparisons w.r.t. data augmentation (R-H, R-G, R-B) are major and should be properly addressed so as to improve the quality of the submission. Authors should justify their choice of a more limited set of rotations for the data augmentation baseline (R-G, R-B) and ideally provide a baseline with matching settings. As suggested by R-3, a clearer definition of sample efficiency in the related work section (e.g. when discussing data augmentation), and experiments that support the claims made in that section---that appear sometimes too strong (R-B), would greatly improve the submission.

I thank the authors for their updates and the new baseline results. The rebuttal addressed most of the reviewers' concern and I thus recommend the paper for acceptance. If possible, in the final version the authors could also discuss the computational complexity, especially since data augmentation alternatives would not increase computational cost at test time.

---

> ### Author Response · Authors · 2021-08-20
> **Author Response to Area Chair afcK**
>
> The authors thank the Area Chair for the meta-review. We are glad that the reviewers found our results encouraging. We have made several changes to the paper to address the issues raised by the reviewers. Please see the following for the details:
>
> The reviewers raised the concern about the fairness regarding the data augmentation (i.e., the "soft equivariant") baselines. Thanks for pointing this out. We made the following changes in the revision:
>
> - We replaced the "soft equivariant" baseline with two more standard image augmentation baselines: RAD [1] and DrQ [2]. We used the same angular discretization in those augmentation methods as we use in the equivariant architecture. Please refer to Section 4.2 and Section 4.3 for the experimental results. Please note that our proposal outperforms the two baselines in 5 of 6 tasks. In Bottle Arrangement exclusively, our method learns faster but converges to a similar level as DrQ.
>
> - We further compared our method against RAD and DrQ with more data augmentation operators, including translation, SE(2), and shift, in Appendix G.1. Note that those data augmentation methods are not directly comparable with our method since those augmentation operators consider other symmetries than rotation. Even so, our proposal outperforms all baseline data augmentation methods in 5 of 6 tasks, except for the bottle arrangement task.
>
> The reviewers also raised a concern about "The Q function architecture beyond the equivariant convolutions not well explained". We interpret this to mean that we should dedicate more space at the beginning of Section 4.3 to explain the ASR method of [29]. We augmented the introduction of ASR in Section 4.3 in the revision.
>
> The reviewers suggested that we should provide a clear definition of sample efficiency. We added that in line 32 in the revision.
>
> The reviewers further raised a concern that some of our claims about the related work are too strong. We revised that claim in line 27 in the revision.
>
> In addition, we reran our simulation box palletizing experiment in simulation to make it consistent with the robot experiment environment. To be more specific, in order to accommodate the fact that we have a different pallet in the real world as we were using in the simulation experiment, in the initial submission, we used a different pallet model and palletized direction in the simulation experiment and the real world experiment (Fig 24 in the initial submission's supplementary appendix). In the revision, we rerun the simulation box palletizing experiment to match the robot experiment environment. Note that this change of environment did not change the performance of either our method or the baselines.
>
> [1] Laskin, Michael, et al. "Reinforcement learning with augmented data." arXiv preprint arXiv:2004.14990 (2020).
> [2] Kostrikov, Ilya, Denis Yarats, and Rob Fergus. "Image augmentation is all you need: Regularizing deep reinforcement learning from pixels." arXiv preprint arXiv:2004.13649 (2020).

---

### Decision · Program_Chairs · 2021-09-13

**Decision:**

Accept (Poster)

**Comment:**

The authors propose in this paper a Q function neural architecture that is invariant to rotations. The consensus among reviewers is the following.

Pros:
- The problem is important and applicable to many tasks.
- The robotics task is well chosen and results are encouraging.

Cons:
- Concerns regarding the fairness of the experimental setup, especially the data augmentation baselines (R-H (Reviewer HHEb), R-G, R-B).
- The Q function architecture beyond the equivarient convolutions not well explained (R-G).

The authors are encouraged to actively engage with the reviewers and revise their submission if they see fit. Concerns regarding the fairness of the comparisons w.r.t. data augmentation (R-H, R-G, R-B) are major and should be properly addressed so as to improve the quality of the submission. Authors should justify their choice of a more limited set of rotations for the data augmentation baseline (R-G, R-B) and ideally provide a baseline with matching settings. As suggested by R-3, a clearer definition of sample efficiency in the related work section (e.g. when discussing data augmentation), and experiments that support the claims made in that section---that appear sometimes too strong (R-B), would greatly improve the submission.

I thank the authors for their updates and the new baseline results. The rebuttal addressed most of the reviewers' concern and I thus recommend the paper for acceptance. If possible, in the final version the authors could also discuss the computational complexity, especially since data augmentation alternatives would not increase computational cost at test time.